# Meiotic Analysis of Gomphidae Species Sheds Light on the Large X Chromosome of the Family (Anisoptera, Odonata)

**Liliana M. Mola** [1,*] , **Pablo J. Rebagliati** [2], **María F. Fourastié** [3] **and Silvia S. Agopian** [4]

1   Laboratorio de Citogenética y Evolución, Departamento de Ecología, Genética y Evolución, Instituto de Ecología, Genética y Evolución (CONICET-UBA), Facultad de Ciencias Exactas y Naturales, Universidad de Buenos Aires, Intendente Güiraldes 2160, Ciudad Autónoma de Buenos Aires C1428EGA, Argentina

2   Facultad de Bromatología, Universidad Nacional de Entre Ríos, Sede Centro. 25 de mayo 709, Gualeguaychú E2822ABA, Entre Ríos, Argentina

3   Instituto de Ecología, Genética y Evolución (CONICET-UBA), Facultad de Ciencias Exactas y Naturales, Universidad de Buenos Aires, Intendente Güiraldes 2160, Ciudad Autónoma de Buenos Aires C1428EGA, Argentina

4   Independent Researcher, Ciudad Autónoma de Buenos Aires C1176ABM, Argentina

*   Correspondence: lilimola@yahoo.com.ar

**Abstract:** In most Anisoptera families, the modal diploid number is 25 in males (24 autosomes + X), and the X chromosome is one of the smallest elements of the complement. The family Gomphidae is an exception, as it has a modal diploid number of 23 (22 + X), and the X chromosome is the largest of the complement and of medium-to-large size in many species. We studied the meiosis of three gomphid species from Argentina: *Aphylla* cf. *distinguenda* (Campion, 1920), *Phyllocycla propinqua* Belle, 1972 and *Phyllocycla* sp. Chromosome number is 2n = 23, n = 11 + X, except for *Phyllocycla propinqua*, showing n = 10 + X. The X chromosome of these species is medium-sized and presents heteropyknotic blocks of different sizes. Despite the small number of gomphid species analysed, there is a clear trend of increasing size of the X chromosome with the increasing amount of heterochromatin. Our results, together with those from the literature, suggest that its large size might have been due to a progressive accumulation of repetitive DNA and heterochromatinisation and not to fusion, as previously suggested. This led us to propose that the ancestral number coincided with the modal number of Gomphidae. A revision of the derived sex-determining systems in Odonata is also provided.

**Keywords:** holokinetic chromosomes; gomphids X chromosome evolution; sex-determination systems; *Aphylla*; *Phyllocycla*

## 1. Introduction

Gomphidae is the second largest family of the suborder Anisoptera, with approximately 1000 species. Among the most species-rich genera, *Aphylla*, *Phyllocycla*, *Phyllogomphoides* and *Progomphus* are mainly distributed in the Neotropics but are also found in the Nearctic region [1].

Odonata exhibits some particular cytogenetic features, such as holokinetic chromosomes (i.e., without primary constriction or centromere) and equatorial division of the X chromosome, but the type of meiosis of autosomes is controversial. Some authors establish that meiosis is post-reductional (i.e., sister chromatids separate in the first division and homologues in the second one). They are based on the orientation of the bivalents with subterminal ciasmata on the equatorial plane, the equational division of the heteromorphic autosomal and neo-sex bivalents, and autosomal trivalents at metaphase I. These lead to the presence of heteromorphic chromatids or three chromatids, respectively, at all metaphases II [2]. Instead, Nokkala and collaborators [3] consider that the meiotic division of the autosomes is pre-reductional (canonical) based on the study of one species that presents

only interstitial chiasmata and their interpretation of the migration of the homologous telomeres at anaphase I. Odonata is also characterised by having a single chiasma in all the bivalents and a noticeably small autosome pair (m-chromosomes) of about half the size of the following pair, which shows a regular meiotic behaviour [2].

The chromosome number has been determined in more than 600 species of the order belonging to 23 families [4]. In the Neotropical region, cytogenetic data have been reported for about 235 species from eight countries (Argentina, Bolivia, Brazil, Chile, Perú, Suriname, Uruguay and Venezuela) [4].

Although the haploid number is relatively constant (12, 13 or 14) in nearly 93% of the species, the chromosome number ranges from n = 3 in *Macrothemis hemichlora* (Burmeister, 1839) to n = 21 in *Orthemis nodiplaga* Karsch, 1891 (Libellulidae). In the families of the suborder Anisoptera, the modal haploid number is 12 in Gomphidae, 13 in Cordulegasteridae, Corduliidae, Libellulidae and Macrodiplacidae, and 14 in Aeshnidae [2,4–6].

About 95% of the species possess an XX/X0 sex-determination chromosome system with male heterogamety. The X chromosome is generally the smallest of the complement or the second smallest element after the m chromosomes. In contrast, it is the largest of the complement in some Gomphidae species and different theories have been postulated to explain its unusual size [7–17]. The sex chromosome systems may have originated from fusions or insertions, such as the neo-XY/neo-XX system and the multiple $X_1X_2Y/X_1X_1X_2X_2$ system in *Micrathyria ungulata* [2,4,6,11,18].

C-banding revealed that most autosomes have heterochromatic blocks in both telomeric regions, which are either small or large and symmetric or asymmetric [19]. The X chromosome of males is entirely C-positive, and exhibits intermediate staining or possesses C-positive bands localised in the terminal or interstitial regions [6,15,19–24].

In the present study, we analysed the meiotic behaviour and the characteristics of the X chromosome in three species of Gomphidae: *Aphylla* cf. *distinguenda* (Campion, 1920), *Phyllocycla propinqua* Belle, 1972 and *Phyllocycla* sp., and discuss the origin of the large X chromosomes and the diploid number of the common ancestor of Gomphidae. Moreover, we provide a review of the derived sex-determining systems in the order.

## 2. Material and Methods

The present study was conducted on three adult males of *Aphylla* cf. *distinguenda* and one adult male of *Phyllocycla* sp. from Tigre in the Lower Delta of the Paraná River (30°28′00″ S 62°49′59″ W) (Buenos Aires Province), and six adult males of *Phyllocycla propinqua* from Arroyo León, Department of Eldorado (26°24′04″ S 54°37′07″ W) (Misiones Province), Argentina. Administration of National Parks of Argentina issued the permit for the collection and transport of material in the protected area.

Within the cosmopolitan family Gomphidae, the genus *Phyllocycla* is distributed from southern Mexico to Uruguay and northern Argentina, and the genus *Aphylla* from southeast United States to Uruguay and northern Argentina [25].

The specimens were etherised in the field, their abdomen was longitudinally incised on the dorsal side and they were whole fixed in 3:1 (absolute ethanol: glacial acetic acid). Later, the gonads were dissected out and immersed in fresh fixative for 24 h before storage in 70% ethanol at 4 °C. For meiotic studies, a piece of gonad was placed in 45% acetic acid for 2 to 3 min to facilitate cell spreading and slides were made by the squash technique in iron propionic haematoxylin.

The chromosome number of *Phyllocycla* sp. and *Aphylla* cf. *distinguenda* was previously communicated in [2], as well as a preliminary study of the meiosis of *Phyllocycla propinqua* was described in [26].

## 3. Results

*Aphylla* cf. *distinguenda* (2n = 23, n = 11 + X). At spermatogonial metaphase there are 23 chromosomes; the X chromosome is the largest of the complement and lies at the centre of an autosomal ring, where the distinguishable pairs of homologues appear to be close to

each other (Figure 1A). During prophase I until diplotene, the large X chromosome shows a large positively heteropyknotic telomeric region, a slim subterminal isopycnotic region and a medium positively heteropyknotic telomeric region (Figure 1B,C). From diplotene onwards, bivalents have one chiasma in submedial or (less frequently) medial position and decrease gradually in size, except for the *m* bivalent, which is half the size of the lower bivalent; the size of the X chromosome is similar to that of the medium bivalents (Figure 1D). At prophase II, the autosomes adopt the typical epsilon ξ-like shape and the X chromosome is composed of a single chromatid (Figure 1E). At metaphase II, the X chromosome lies on the equatorial plate together with the autosomes (Figure 1F).

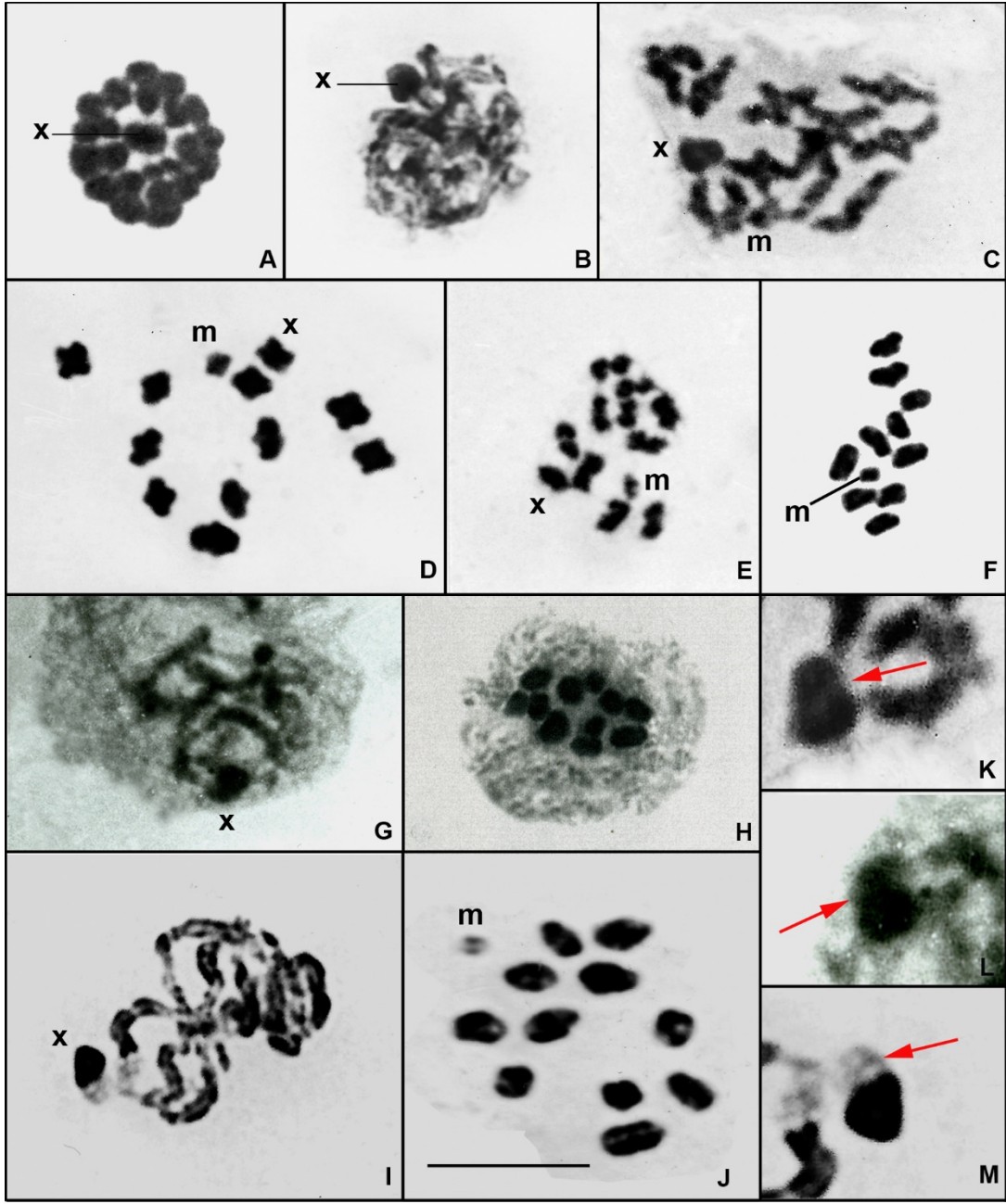

**Figure 1.** *Aphylla* cf. *distinguenda* (2n = 23, n = 11+ X) (**A–F,K**), *Phyllocycla propinqua* (n = 10 + X) (**G,H,L**) and *Phyllocycla* sp. (n = 11 + X) (**I,J,M**). A—Spermatogonial prometaphase, **B**—Pachytene, **C**—Diplotene, **D**—Diakinesis, **E**—Prophase II, **F**—Metaphase II with n = 11, **G**—Pachytene, **H**—Metaphase I, **I**—Late pachytene, **J**—Prometaphase I, **K–M**—Magnifications of X chromosomes from figures **C, G, I, respectively**. Arrows point isopycnotic regions. Bar: **A–J** = 10 um, **K–M** = 5 um.

One of the individuals studied showed a variation in the chromosome number at the first and second meiotic divisions. Of 28 diplotenes and diakineses analysed, 23 cells presented 11 + X and 5 10 + X, whereas of 27 prometaphases and metaphases II, 2 presented 11 + X and 25 10 + X (Figure 1E,F).

*Phyllocycla propinqua* (n = 10 + X). At pachytene, the X chromosome is large and shows a large positively heteropyknotic telomeric region, a slim subterminal isopycnotic region and a small positively heteropyknotic telomeric region similar to *Aphylla distinguenda* (Figure 1G). At diakinesis and subsequent meiotic stages, the sex chromosome becomes isopycnotic with the bivalents and its size is similar to that of the median bivalents (Figure 1H). Bivalents show a terminal chiasma, and a slightly larger bivalent can be recognized, whereas the remaining ones decrease gradually and the m chromosomes are absent (Figure 1H). All prometaphases show 11 chromosomes, at metaphase II, no chromosome is observed out of the equatorial plate, and at anaphase II, all chromosomes migrate synchronously.

*Phyllocycla* sp. (n = 11 + X). From the early prophase II onwards, the X chromosome is large and positively heteropycnotic (Figure 1I). At pachytene the X chromosome has a large positively heteropycnotic region and a small terminal isopycnotic region (Figure 1I). Bivalents possess a single terminal chiasma and decrease gradually in size, except for the bivalent formed by the small m chromosomes, which are negatively heteropycnotic (Figure 1). At diakinesis, the X chromosome becomes isopycnotic and its size is similar to that of the median bivalents; it is hardly recognisable at prometaphase and metaphase I.

## 4. Discussion

Cytogenetic studies conducted on 76 species of Odonata revealed a modal number of n = 12 (11 + X) (Table 1). A reduction in modal number through fusions has been reported in five species, resulting in haploid numbers of 10 + X and 9 + X (Table 1). Interpopulation variation in chromosome number has been recorded only in two species (*Asiagomphus melaenops* and *Trigomphus melampus*), probably due to species misidentification.

**Table 1.** Chromosomal data by Gomphidae species.

| | Species | n (Male) | X Size in Mitosis | X Size in Meiosis | Locality | References |
|---|---|---|---|---|---|---|
| 1 | *Anisogomphus bivittatus* (Selys, 1854) | 11 + X<br>11 + X<br>11 + X | -<br>LL<br>- | S<br>S<br>S | India<br>India<br>Nepal | [27]<br>[28]<br>[29] as *Temnogomphus bivittatus* (Selys, 1854) |
| 2 | *A. occipitalis* (Selys, 1854) | 11 + X | - | S<br>- | Nepal<br>India | [29]<br>[30] |
| 3 | *Aphylla* cf. *distinguenda* (Campion, 1920) | 11 + X | LL | M | Argentina | [2] this work |
| 4 | *A. edentata* Selys, 1869 | 11 + X | - | - | Bolivia | [5] |
| 5 | *A. producta* Selys, 1854 | 11 + X | - | - | Bolivia | [5] |
| 6 | *A. theodorina* (Navas, 1933) | 11 + X | LL | LL | Brazil | [13] |
| 7 | *A. williamsoni* (Gloyd, 1936) | 11 + X | M | M | USA | [31] |
| 8 | *Arigomphus lentulus* (Needham, 1902) | 11 + X | - | - | USA | [32] as *Gomphus lentulus* Needham, 1902 |
| 9 | *A. pallidus* (Rambur, 1842) | 11 + X | - | - | USA | [5] as *Gomphus pallidus* Rambur, 1842 |

**Table 1.** *Cont.*

| | Species | n (Male) | X Size in Mitosis | X Size in Meiosis | Locality | References |
|---|---|---|---|---|---|---|
| 10 | *A. submedianus* (Williamson, 1914) | 11 + X | - | - | USA | [32] as Gomphus submedianus Williamson, 1914 |
| 11 | *Asiagomphus melaenops* (Selys, 1854) | 9 + X <br> 9 + X <br> 11 + X <br> 11 + X | - <br> - <br> LL <br> LL | - <br> M <br> L <br> L | Japan <br> Japan <br> Japan <br> Japan | [33] <br> [34] <br> [35] <br> [14] all as *Gomphus melaenops* Selys, 1854 |
| 12 | *Burmagomphus* cf. *arboreus* Lieftinck, 1940 | 11 + X | - | - | India | [30] |
| 13 | *B. divaricatus* Lieftinck, 1964 | 11 + X | - | M | India | [36] |
| 14 | *B. pyramidalis* Laidlaw, 1922 | 11 + X <br> 11 + X | - | S-M <br> M | India <br> India | [30,37] <br> [36] |
| 15 | *B. sivalikensis* Laidlaw, 1922 | 11 + X | - | M | India | [36] |
| 16 | *B. williamsoni* Förster, 1914 | 11 + X | - | M | India | [36] |
| 17 | *Davidius fujiama* Fraser, 1936 | 11 + X <br> 2n = 24 F | H | LL | Japan | [15] |
| 18 | *D. moiwanus* (Okumura, 1935) | 11+X | H | M | Japan | [15] as *D. m. moiwanus* (Okumura) |
| 19 | *D. nanus* (Selys, 1869) | 11 + X <br><br> 11 + X <br> 11 + X | - <br><br> - <br> M | S <br><br> M <br> M | Japan <br><br> Japan <br> Japan | [38] as *Gomphus hakiensis* Oguma, 1926 <br> [14] <br> [16] |
| 20 | *Dromogomphus spinosus* (Selys, 1854) | 11 + X | - | - | USA | [32] |
| 21 | *D. spoliatus* (Hagen, 1857) | 11 + X | - | - | USA | [32] |
| 22 | *Epigomphus llama* Calvert, 1903 | 9 + X | - | - | Bolivia | [5] |
| 23 | *Erpetogomphus designatus* Hagen, 1857 | 11 + X | - | - | USA | [5] |
| 24 | *E. diadophis* Calvert, 1905 | 11 + X | - | - | USA | [5] |
| 25 | *E. ophibolus* Calvert, 1905 | 11 + X | - | M | Mexico | [39] |
| 26 | *Gomphoides* sp. | 11 + X | - | - | Bolivia | [5] |
| 27 | *Gomphus confraternus* Selys, 1873 | 11 + X | - | - | USA | [32] |
| 28 | *G. exilis* Selys, 1854 | 11 + X <br> 2n = 24F | - | - <br> - | USA <br> Canada | [32] <br> [11] |
| 29 | *G. graslini* Rambur, 1842 | 11 + X | LL | - | France | [10] [11] |
| 30 | *G. pulchellus* Selys, 1840 | 11 + X | - | M-L | France | [40] |
| 31 | *G. vulgatissimus* (Linnaeus, 1758) | 11 + X | - | s | Russia | [20] |
| 32 | *Ictinogomphus decoratus* (Selys, 1854) | 11 + X | LL | M-L | Singapur | [41] as *I. decoratus melaenops* |

**Table 1.** *Cont.*

| | Species | n (Male) | X Size in Mitosis | X Size in Meiosis | Locality | References |
|---|---|---|---|---|---|---|
| 33 | *I. pertinax* (Hagen in Selys, 1854) | 11 + X | - | M | Japan | [14] |
| 34 | *I. rapax* (Rambur, 1942) | 11 + X<br>11 + X<br>11 + X | H<br>-<br>LL | LL<br>-<br>LL | India<br>India<br>India | [7] as *Ictinus rapax*<br>[42–45]<br>[9] as *Ictinus rapax*<br>Omura 1949 |
| 35 | *Nepogomphus modestus* (Selys, 1878) | 11 + X<br>11 + X | -<br>M | M<br>M | India<br>India | [46]<br>[22] |
| 36 | *Nihonogomphus ruptus* (Selys, 1858) | 11 + X | - | S | Russia | [20] |
| 37 | *N. viridis* Oguma, 1926 | 11 + X<br>11 + X | -<br>H | L<br>L | Japan<br>Japan | [35]<br>[16] |
| 38 | *Octogomphus specularis* (Hagen, 1859) | 11 + X | - | - | USA | [32] |
| 39 | *Onychogomphus forcipatus* (Linnaeus, 1758) | 11 + neo-XY | | | Austria | [11] |
| 40 | *O. saundersii* Selys, 1854 | 11 + neo-XY | | | India | [12,30,37,47]<br>as *O. s. duaricus* Fraser,<br>1924 |
| 41 | *O schmidti* Fraser, 1937 | 11 + neo-XY | | | India | [12,30,47,48] |
| 42 | *Ophiogomphus bison* Selys, 1873 | 11 + X/12 + X | - | - | USA | [32] |
| 43 | *O. cecilia* (Fourcroy, 1785) | 12 F<br><br><br>11 + X<br>11 + X | -<br><br><br>-<br>H | LL<br><br><br>LL<br>LL | Finland<br><br><br>Russia<br>Russia | [8] as *O. serpentinus*<br>Charp.<br>Syn *Aeschna serpentina*<br>Charpentier, 1825<br>[19]<br>[17] as *O. c. cecilia*<br>(Four.) |
| 44 | *O. colubrinus* Selys, 1854 | 11 + X | - | - | USA | [32] |
| 45 | *O. obscurus* Bartenev, 1909 | 11 + X | - | - | Russia | [49] |
| 46 | *O. occidentalis* Hagen, 1882 | 11 + X | - | - | USA | [32] |
| 47 | *O. rupinsulensis* (Walsh, 1862) | 11 + X | - | S-M | USA | [32] |
| 48 | *O spinicornis* Selys, 1878 | 11 + X | - | LL | China | [50] as *O. spinicorne* |
| 49 | *Paragomphus capricornis* (Förster, 1914) | 11 + X | - | L | Thailand | [51] |
| 50 | *P. lineatus* (Selys, 1850) | 11 + X<br>11 + X<br>11 + X | -<br>-<br>L | M<br>-<br>M | Nepal<br>India<br>India | [29]<br>[30]<br>[22] |
| 51 | *Phanogomphus lividus* (Selys, 1854) | 11 + X | - | - | USA | [32] as *Gomphus lividus*<br>Selys,<br>1854 |
| 52 | *Ph. militaris* (Hagen, 1858) | 11 + X | - | - | USA | [32] as *Gomphus*<br>*militaris*<br>Hagen, 1858 |

**Table 1.** *Cont.*

| | Species | n (Male) | X Size in Mitosis | X Size in Meiosis | Locality | References |
|---|---|---|---|---|---|---|
| 53 | *Ph. spicatus* (Selys, 1854) | 11 + X | - | - | USA | [32] as *Gomphus spicatus* Selys, 1854 |
| 54 | *Phyllocycla propinqua* Belle, 1972 | 10 + X | - | M | Argentina | [51] this work |
| 55 | *Phyllocycla* sp. | 11 + X | - | - | Bolivia | [5] |
| 56 | *Phyllocycla* sp. | 11 | - | - | Brazil | [52] |
| 57 | *Phyllocycla* sp. | 11 + X | - | M | Argentina | [2] this work |
| 58 | *Phyllogomphoides undulatus* (Needham, 1944) | 11 + X | - | S | Surinam | [53] |
| 59 | *Progomphus borealis* McLachlan, 1873 | 11 + X | - | - | USA | [32] |
| 60 | *P. intricatus* (Hagen, 1857) | 11 + X<br>11 + neo-XY | -<br>- | -<br>- | Bolivia<br>Brazil | [5]<br>[52] |
| 61 | *P. obscurus* (Rambur, 1842) | 11 + X | - | - | USA | [32] |
| 62 | *P. phyllochromus* Ris, 1918 | 11 + X | - | - | Bolivia | [5] |
| 63 | *Scalmogomphus bistrigatus* (Hagen, 1854) | 11 + X<br>11 + X | -<br>- | LL<br>LL | Nepal<br>India | [29]<br>[30,54] both as *Onychogomphus bistrigatus* (Hagen, 1854) |
| 64 | *Shaogomphus postocularis* (Selys, 1869) | 11 + X<br><br>11 + X | -<br><br>- | -<br><br>S | Japan<br><br>Russia | [16,35] both as *Gomphus postocularis* Selys, 1869<br>[20] as *Gomphus epophtalmus* Selys, 1872 |
| 65 | *Sieboldius albardae* Selys, 1886 | 11 + X<br>11 + X<br>11 + X | -<br>H<br>H | LL<br>LL<br>LL | Japan<br>Japan<br>Japan | [35]<br>[14]<br>[16] |
| 66 | *Sinictinogomphus clavatus* (Fabricius, 1775) | 11 + X | LL | M | Japan | [14] as *Ictinogomphus clavatus* (Fabricius, 1775) |
| 67 | *Stylogomphus suzukii* (Oguma, 1926) | 11 + X | -<br>- | S<br>- | Japan<br>Japan | [55]<br>[42] both as *Gomphus suzukii* Oguma, 1926 |
| 68 | *Stylurus flavipes* (Charpentier, 1825) | 11 + X | - | - | Russia | [48] |
| 69 | *S. plagiatus* (Selys, 1854) | 11 + X | - | - | USA | [32] as *Gomphus plagiatus* Selys, 1854 |
| 70 | *S. scudderi* (Selys, 1873) | 11 + X | - | - | USA | [32] as *Gomphus scudderi* Selys, 1873 |

**Table 1.** *Cont.*

| | Species | n (Male) | X Size in Mitosis | X Size in Meiosis | Locality | References |
|---|---|---|---|---|---|---|
| 71 | *S. townesi* Gloyd, 1936 | 11 + neo-XY | | | USA | [31] as *Gomphus townesi* Gloyd, 1936 |
| 72 | *Trigomphus citimus* (Needham, 1931) | 10 + X<br>10 + X<br>10 + X | -<br>-<br>- | -<br>S<br>L | Japan<br>Japan<br>Japan | [33]<br>[34]<br>[14] all as *Gomphus citimus tabei* Asahina, 1949 |
| 73 | *T. interruptus* (Selys, 1854) | 9 + X | - | L | Japan | [14] |
| 74 | *T. melampus* (Selys, 1869) | 9 + X<br><br>10 + X<br><br>9 + X<br><br>11 + X<br><br>10 + X | -<br><br>-<br><br>-<br><br>-<br><br>LL | M<br><br>L<br><br>M-L<br><br>S<br><br>M-S | Japan<br><br>Japan<br><br>Japan<br><br>Japan<br><br>Japan | [55] as *Gomphus melampus* Selys, 1869<br>[55] as *Gomphus unifasciatus* Oguma, 1926<br>[35] as *Gomphus melampus bifasciatus* Asahina<br>[34] as *Gomphus m. bifasciatus* Asahina<br>[16] |
| 75 | *T. ogumai* Asahina, 1949 | 10 + X | - | s | Japan | [14] |
| 76 | *Zonophora callipus* Selys, 1869 | 11 + X | LL | M | Surinam | [53] |

**Notes: s**—the smallest chromosome, **S**—among small chromosomes/bivalents, **M**—among medium chromosomes/bivalents, **L**—among large chromosomes/bivalents, **LL**—the largest chromosome/bivalent, **H**—huge chromosome. This table is adapted from Table 1, Gomphidae of [4], with information regarding the chromosome X length and with the addition of data of new bibliography.

Five of the 24 species of *Aphylla* so far described have been studied cytogenetically, showing n = 11 + X (Table 1) [1]. In *A. theodorina*, *A.* cf. *distinguenda* (analysed here), *Ictinogomphus decoratus*, *I. rapax* and *Zonophora callipus*, the median to large X chromosome is located in the centre of the mitotic metaphase plate [7,13,41,53]. This is an unusual feature as in most species with small X chromosomes, it does not adopt any particular arrangement at mitosis.

An intraindividual variation in chromosome number was detected in a specimen of *Aphylla* cf. *distinguenda*. This may be related to the fact that the testes of odonates consist of globular cysts arranged around a central duct running the length of each gonad [56,57]. A spermatogenic wave begins at a given point and then proceeds slowly towards the rest of the gonad, so that most cells in each cyst are at the same developmental stage. On this basis, the difference in the chromosome number found in the individual of *Aphylla* cf. *distinguenda* was probably due to the abnormal segregation of a bivalent at meiosis I, leading to its loss in the cell or cells that gave rise to the cysts with 10 + X analysed. Such reduction would be represented at the second meiotic division. An increase in the proportion of cells with a missing chromosome at meiosis II may be the result of a sampling error, but this is unlikely because we analysed the same number of cells at diakinesis. Another possible explanation is that individual cysts were differently affected by different environmental conditions, and that the most represented cysts were those missing one chromosome at meiosis II.

In *Phyllocycla*, cytogenetic studies have been performed in South American specimens from five of the 31 species so far described (Table 1) [1]. The modal number of the family is present in *Phyllocycla* sp. studied here and in *Phyllocycla* sp. studied by [5]. Souza Bueno [52] suggested a sex-determining mechanism other than X for a specimen of *Phyllocycla* sp. (n = 11), but it cannot be identified because the diploid number was not reported. On the other hand, *Phyllocycla propinqua* (n = 10 + X) shows a reduction in the number of autosomes.

This was probably due to autosomal fusion in homozygous condition, as suggested by the presence of a larger bivalent not observed in other species.

Another distinct feature of the species analysed cytogenetically is the spatial arrangement of the large X chromosome at meiosis II, as it is aligned on the equatorial plate at metaphase II and migrates synchronously with the autosomes at anaphase II. In contrast, in species with small X chromosome it is usually outside the metaphase plate at metaphase II and migrates asynchronously with the autosomes (generally lying ahead) at anaphase II.

### 4.1. Ancestral Chromosome Number of Gomphidae

Kiauta [11] assumed that the ancestral number of the family was 2n = 25 and that the process of reduction in chromosome number occurred in three successive steps, all of which involved the sex chromosome. First, the X chromosome might have fused to an autosome (A), giving rise to the neo-XY system and reducing the chromosome number to 24. In the second step, the neo-Y (A') chromosome probably fused to another autosome (B); this resulted in a neo-neo-Y chromosome and a neo-X/neo-neo-Y system, with the consequent reduction in the chromosome number to 23. In the third step, the autosomal portion (A) of the neo-X might have been translocated to the autosomal homologue (B') (originating a chromosome homologous to the neo-neo-Y), thereby restoring the X0 system without change in the chromosome number (2n = 23). It is worth mentioning that the X chromosome most likely retained its original small size after these arrangements.

According to [11], the first two steps were supported by cytogenetic evidence. With regard to the third step for which no evidence was available, the author argued that it was necessary for originating the chromosome complement found in most species of Gomphidae with a detached X chromosome. Kiauta [11] cited *Onychogomphus forcipatus* as the primary example to support the hypothesis of chromosome number reduction from 25 to 24. The author assumed that the variation in chromosome number (12–13 elements) observed among meiotic cells of a same specimen was due to a "reversible fusion" of the X chromosome, that is to say that it was fused to an autosome in some cells and unfused in others. On the other hand, Mola [58] performed the cytological analysis of *Rhionaeschna bonariensis* (12 + neo-XY) and confirmed that such karyotypic variation was due to the presence of univalents derived from desynapsis of the sex bivalent. This explanation could also account for the karyotypic variation in *Onychogomphus forcipatus*. To support the complement reduction from 24 to 23 chromosomes, Kiauta [11] hypothesised that the X chromosome of *Gomphus graslini*, *Ophiogomphus cecilia* and *Ictinogomphus rapax* is large because it was formed by fusion, thereby being a neo-X. Taking into account the rearrangements proposed by this author, at meiosis are expected to observe different configurations. The first configuration is a trivalent formed by the pairing of the neo-neo-Y with the autosome B' and the neo X. If chiasma formation fails, there are two different configurations. One of them is a heteromorphic bivalent formed by the pairing of the neo-neo-Y with the autosome B' and a large univalent corresponding to the neo-X. The other is a heteromorphic bivalent formed by the pairing of the neo-neo-Y with the neo-X and a univalent corresponding to the autosome B'. However, *Ictinogomphus rapax* and *Ophiogomphus cecilia* show 11 homomorphic bivalents and the large X chromosome at meiosis [7,9,17,19]. Only the characteristics of the 23 mitotic chromosomes are available for *Gomphus graslini* [11].

Tyagi [12,47] also conducted evolutionary studies of the karyotype of Gomphidae based on species of *Onychogomphus*. This author included the neo-neo-X/neo-neo-Y system to the scheme proposed by Kiauta [11] and suggested that it resulted from the neo-X/neo-neo-Y through the fusion of an autosome with the neo-X, giving rise to the neo-neo-X. The existence of this neo-system was strongly suggested by the finding of a reduced chromosome number in some spermatogonial cells, and the variation in the chromosome number of different meiotic cells was assumed to derive from "unstable fusions" (i.e., "reversible fusions").

Kiauta [11] and Tyagi [47] supported the hypothesis that the ancestral chromosome number of Gomphidae was 25 mainly due to the presence of some species with a very large X chromosome and other species with a complement of 2n = 24 and a neo-XY sex-determination system. However, the theoretical steps are not reflected in the course of meiosis and some of their observations can be interpreted from a different perspective (as mentioned above). In this context, the origins of the large X chromosome and the neo-XY sex-determination system are discussed below.

Later, Perepelov and Bugrov [17] proposed that the large X chromosome of *Ophiogomphus cecilia* originated from the fusion of the original X chromosome with two chromosomes of the same autosomal pair. This hypothesis seems unlikely as it would result in a genetic imbalance in both sexes: in females by the duplication of one chromosome pair and in males by the duplication of one chromosome.

Considering the discussion presented above, together with the fact that the modal chromosome number of the family is 23 (n = 11 + X) in about 86% species, we propose that the ancestral chromosome number of Gomphidae coincides with the modal chromosome number of the family.

### 4.2. Characterisation of the Large X Chromosome of Gomphidae

The size of the sex chromosome has been described or illustrated at the mitosis or meiosis of about 60% of Gomphidae species (Table 1). In most odonates, the X chromosome is the smallest or the second smallest of the complement if the m chromosomes are present. In Gomphidae, the X chromosome is the smallest of the complement in only two species (*Gomphus vulgatissimus* and *Trigomphus ogumai*) [14,20]. It may be the size of the smaller bivalents [20,27–29,34,38,53,55] or the largest of the complement [7–9,13–17,19,29,30,35, 50,54], but in most cases its size is similar to that of the medium and large bivalents (Table 1). In mitosis, it may even be considerably larger (huge chromosome) than the other chromosomes of the complement (Table 1).

The distribution of the constitutive heterochromatin in the autosomes of gomphids matches with that of the other families analysed cytogenetically. The heterochromatin is found in the telomeric region of all the chromosomes (except for *Nepogomphus modestus*, with a bivalent lacking C-bands), and the number of repeats may vary among species, chromosomes or different telomeric regions of a same chromosome [6,17,20,22,28,36].

The X chromosome in species of other families with C-banding may be entirely C-positive, may show small C-terminal bands on one or both telomeric regions or may be isopycnotic [6]. Although the X chromosome is entirely C-positive in about half of the species analysed, this is possibly due to a higher degree of contraction (facultative heterochromatinisation in males) rather than to the presence of constitutive heterochromatin.

In Gomphidae, the amount and distribution of constitutive heterochromatin on the X chromosome have provided evidence to explain the origin of its large size. The X chromosome of *Gomphus vulgatissimus* is the smallest of the complement and has a low amount of heterochromatin in both telomeric regions [20]. In *Shaogomphus postocularis* the X chromosome presents a large terminal heterochromatic region covering about half of the chromosome and in *Nihonogomphus ruptus* it has large terminal heterochromatic blocks in both telomeric regions. In these two species, the size of the X chromosome is similar to that of the smaller bivalents [20]. In *Paragomphus lineatus*, the X chromosome is almost entirely heterochromatic and shows an euchromatic submedial region, and in *Nepogomphus modestus* it contains a large heterochromatic region and a small euchromatic segment in terminal position. In both species the X chromosome is of similar size to that of medium bivalents [22]. In *Ophiogomphus cecilia* the sex chromosome contains a large inhomogeneous heterochromatic portion and an euchromatic one with three intercalar heterochromatic segments and in *Davidius fujiama* it presents three large heterochromatic blocks, two of which are located in terminal position. In these two species the X chromosome is the largest of the complement. Although the other two species of *Davidius* were not analysed by C-banding, in meiosis their X chromosome exhibits three darker areas similar to those

present in *D. fujiama* [15–17]. Intensely stained blocks can be frequently distinguished in preparations with no banding patterns, which may correspond to C bands [6].

Therefore, the heteropycnotic regions observed at prophase I in the X chromosome of *Aphylla* cf. *distinguenda*, *P. propinqua* and *Phyllocycla* sp. analysed here could be considered as regions of constitutive heterochromatin.

Despite the fact that the heterochromatin of the X chromosome has been studied in a small number of gomphid species, there is a clear trend towards an increase in heterochromatin amount with increasing X chromosome size. This allows us to propose that its large size would have originated by progressive accumulation of repetitive DNAs and heterochromatinisation rather than to fusions, as previously suggested [11,12,17,47].

### 4.3. Derived Sex-Determining Systems

In Odonata, no reports have been published on sex-determining systems originated by fragmentation of the X chromosome, as documented for other insects with holokinetic chromosomes such as Heteroptera and Lepidoptera [59].

In Odonata families with a small X chromosome (except Gomphidae), the identification of a heteromorphic sex bivalent at the different meiotic stages may be a difficult task, depending on both the size of the autosome with which it might have fused and on the degree of contraction of the bivalents. In about half of the species with the neo- XY system, the sex bivalent in males is homomorphic throughout meiosis and its presence can be mainly inferred from the even number of chromosomes in spermatogonial cells (Table 2). On the contrary, in the other species the sex bivalent is heteromorphic in meiosis I and II, or it is recognised only at diplotene and diakinesis (Table 2).

**Table 2.** Chromosomal data of species, subspecies, populations or individuals with derived sex determination systems.

| Family Species | Suborder | 2n Male | n Male | H | SBS | N | Locality | References |
|---|---|---|---|---|---|---|---|---|
| | Anisoptera | | | | | | | |
| **Aeshnidae** | | | | | | | | |
| *Aeshna caerulea* (Ström, 1783) | | | 11 + neo-XY | Y | SS | 2 | Finland | [60] |
| | | 27 | 13 + X | | | - | USSR | [61] |
| | | 25 | 12 + X | | | - | USSR | [62] |
| | | 25 | X | | | - | Finland | [20] |
| *A. grandis* (Linnaeus, 1758) | | 26 | 12 + neo-XY | Y | LL | 23 | Finland | [8,60] |
| | | 26F | 12 + neo-XX | | | - | | |
| | | | 12 + neo-XY | Y | LL | 8 | Netherlands | [10,11,63] |
| | | 26 | 12 + neo-XY | Y | LL | - | Russia | [21] |
| | | | 12 + X | | | - | USSR | [62] |
| | | 26 | 12 + neo-XY | N | - | 14 | Finland | [60] |
| *A. juncea* (Linnaeus, 1758) | | 26F | | - | - | 3 | | |
| | | | 12 + neo-XY | D | L | 6 | Italy | [57] |
| | | 26 | 12 + neo-XY | Y ‡ | L | 4 | Russia | [21] |
| *A.serrata* Hagen, 1856 | | | 12 + neo-XY | N | - | 1 | Finland | [60] as *A.osiliensis fennica* |

**Table 2.** *Cont.*

| Family Suborder Species | 2n Male | n Male | H | SBS | N | Locality | References |
|---|---|---|---|---|---|---|---|
| *A. viridis* Eversmann, 1836 | - | 12 + neo-XY | N | - | 3 | Finland | [60] |
| | 26F | | - | - | 2 | Russia | [19] |
| | 26 | 12 + neo-XY | D ‡ | LL | 2 | | |
| *Anax ephippiger* (Burmeister, 1839) | 14 14F | 6 + neo-XY | D | M | - | India | [64] as *Hemianax ephippiger* |
| *Caliaeschna microstigmata* (Schneider, 1845) | | 6 + neo-XY | N | - | 1 | Greece | [65] |
| *Gynacanta interioris* Williamson, 1923 | 26 | 12 + neo-XY | D | M | 2 | Brazil | [13] |
| *Rhionaeschna bonariensis* (Rambur, 1842) | 26 | 12 + neo-XY | Y | LL | 5 | Argentina | [58,66] both as *Aeschna bonariensis* |
| | | | Y | LL | 2 | Uruguay | |
| *R. planaltica* (Calvert, 1845) | 16 | 7 + neo-XY | Y | SS | 2 | Argentina | [58,66] both as *Aeschna cornigera planaltica* |
| **Gomphidae** | | | | | | | |
| *Onychogomphus forcipatus* (Linnaeus, 1758) | 24 | 11 + neo-XY | Y | LL | - | Austria | [11] |
| † *O. saundersii* Selys, 1854 | 22? 23? | 11 + neo-XY | Y | LL | 10 | India | [37] as *O. saundersi duaricus* |
| *O. schmidti* Fraser, 1937 | 22 | 11 + neo-XY | - | LL | - | India | [47,48] |
| *Progomphus intricatus* (Hagen in Selys, 1858) | 23 | 11 + X | - | - | - | Bolivia | [5] |
| | 24 | 11 + neo-XY | N | - | 3 | Brazil | [52] |
| *Stylurus townesi* Gloyd, 1936 | 23? | 11 + neo-XY | Y | LL | 1 | USA | [31] as *Gomphus townesi* |
| **Libellulidae** | | | | | | | |
| *Crocothemis servilia* (Hagen, 1857) *servilia* | 25 | 12 + X | | | | India, Nepal, China, Philippines, Japan Singapore, Korea, Thailand | [7,29,30,42,43,67–75] |
| *C. servilia mariannae* ssp. n. | 24 | 11 + neo-XY | N | - | 3 | Japan | [76] |
| | | 11 + neo-XY | D | S | 5 | Japan | [69] |
| | | 11 + neo-XY | Y | SS | - | Japan | [74] |
| | | 11 + neo-XY | Y | SS | 25 | Japan | [75] |
| *Elasmothemis williamsoni* (Ris, 1919) | 22 | 11 + neo-XY | N | - | 2 | Surinam | [53] as *Dythemis williamsoni* |
| *Erythrodiplax media* Borror, 1942 | | 10 + X | | | - | Bolivia, Brazil | [5,13,52] |
| | 22F | 11 | | | 1 | Brazil | [77] |
| | 22 | 10 + neo-XY | D | L | 8 | Argentina | [78] |
| *Macrothemis hemichlora* (Burmeister,1839) | 6 | 2 + neo-XY | N | - | - | Bolivia | [5] |

**Table 2.** *Cont.*

| Family Species | Suborder | 2n Male | n Male | H | SBS | N | Locality | References |
|---|---|---|---|---|---|---|---|---|
| *Micrathyria longifasciata* Calvert, 1909 | | 24 | 11 + neo-XY | Y | LL | 8 | Argentina | [79] |
| *M. ungulata* Foerster, 1907 | | 23 | **10 + X$_1$X$_2$Y** | Y | M | 2 | Argentina | [18] |
| *Neurothemis tulia* (Drury, 1773) | | 28 | 13 + neo-XY<br>12 + X | Y | U | -<br>4 | India<br>Thailand | [54,67]<br>[73] all *as N.t.tulia* |
| *Orthemis aequilibris* Calvert, 1909 | | 12 | 5 + neo-XY | N | - | 1 | Surinam | [53] |
| *O.ambinigra Calvert, 1909* | | 12 | 5 + neo-XY | N | - | 19 | Argentina | [2,6,80] |
| *O. discolor* (Burmeister, 1839) | | 23<br><br>25<br>23<br><br>24 F | 11 + X<br>11 + neo-XY<br>10 + neo-XY<br>11 + X | | | 4 | Surinam<br><br>Perú, Surinam, Brazil, Argentina<br>Argentina | [53] as *O. ferruginea*<br><br>[2,13,52,53,77] all as *O. ferruginea*<br>[6] |
| O. levis Calvert, 1906 | | 7 | **2II + III** | N | - | 2 | Bolivia | [5] |
| *Orthemis* sp. | | 10 | 4 + neo-XY | N | - | 4 | Bolivia | [5] as *O. ferrugínea* |
| *Pseudothemis zonata* (Burmeister, 1839) | | - | 11 + neo-XY | Y | LL | 6 | Japan | [76] |
| *Trithemis aurora* (Burmeister, 1839) | | 25 | 12 + X<br>9 + neo-XY | –<br>- | –<br>- | –<br>4 | India<br>Nepal | [30,81]<br>[29] |
| **Corduliidae** | | | | | | | | |
| *Somatochlora borisi* Marinov, 2001 | | | 10 + neo-XY | N | - | 7 | Bulgaria | [82] |
| **Zygoptera** | | | | | | | | |
| **Coenagrionidae** | | | | | | | | |
| *Ischnura lobata* Needham, 1930 | | | 13 + neo-XY | Y | LL | 5 | China | [50] |
| *Leptagrion macrurum* (Burmeister, 1839) | | 30 | neo-XY | - | - | 2 | Brazil | [83] |
| *Mecistogaster* sp.2 | | 12 | 5 + neo-XY | N | - | - | Bolivia | [5] |
| **Lestidae** | | | | | | | | |
| *Lestes vigilax* Selys, 1862 | | | **9II + III** | - | - | 1 | USA | [31] |
| **Megapodagrionidae** | | | | | | | | |
| *Heteragrion* sp. b | | 26 | 12 + neo-XY | D | M | 2 | Brazil | [52] |

Notes: **H:** heteromorphism of the sex bivalent: Y yes, N no, D until Diakinesis. ‡: With C Banding. **SBS:** Sex bivalent size: LL the largest of the complement, L among large bivalents, M among medium bivalents, S among small bivalents, SS the smallest of the complement, U sex chromosomes as univalents in all meiotic stages. **N:** number of individuals analysed. **F:** female. **II:** bivalent. **III:** trivalent. **†:** *O. saundersii* Selys, 1854 or *Nychogomphus duaricus* (Fraser, 1924). Both names were assigned by Kuznetsova and Golub [4] for the species studied by Tyagi [37] as *O. saundersi duaricus*.

Neo-systems are rare, as they have been recorded in 35 species, subspecies, populations or some individuals from a population of a total of more than 600 cytogenetically analysed

species of damselflies and dragonflies [4] (Table 2). They have a heterogeneous distribution and are found in only 5 species of three families of Zygoptera and in 30 species of four families of Anisoptera (Table 2). Some genera stand out by the presence of neo-systems in over 25% of their analysed species, such as *Aeshna, Rhionaeschna, Onychogomphus* and *Orthemis* [4] (Table 2).

In Gomphidae, the presence of the neo XY system is always associated with an increase in the modal diploid number of males. The increase in the diploid number may be due to an autosomal fragmentation that in homozygous condition would increase the diploid number to 25. The origin of the neo XY system is due to the fusion of the X chromosome with an autosome that would reduce the diploid number to 24. These two rearrangements may be independent, which is to say that the X chromosome does not necessarily have to fuse with one of the fragmented autosomes. The sex bivalent is the largest of the complement in the three species of *Onychogomphus* and in *Stylurus townesi*; suggesting that the fusion involved the X chromosome and the largest autosome [11,31,37,47,48]. The fact that the three species of *Onychogomphus* studied exhibit the same autosomal pair involved in the fusion may indicate that this rearrangement occurred in a common ancestor. In *Stylurus townesi* the fusion most likely had an independent origin since other three species of the genus have the modal haploid number (11 + X) of the family (Table 1). The presence of a neo-XY system in individuals of *Progomphus intricatus* from Brazil was postulated on the basis of the diploid number; however, its presence could not be detected at meiosis [52]. The occurrence of a chromosome number higher than the modal number of the family together with a neo-XY system has been reported not only for Gomphidae but also for *Neurothemis tulia* (Libellulidae); *Ischnura lobata*; *Leptagrion macrurum* (Coenagrionidae) and *Heteragrion* sp. b (Megapodagrionidae) [50,52,54,67,83].

In seven other species, the presence of the neo-XY system is associated with a remarkable reduction in the entire chromosome complement (diploid numbers between 6 and 16) (Table 2). However, the existence of some species (e.g., *Perithemis lais* (Perty, 1834), *Anax guttatus* (Burmeister, 1839) and *Rhionaeschna intricata* (Martin, 1908)) with reduced complements not involving the sex chromosome may indicate that it is not predisposed towards fusion [5,13,68].

Summarizing, the neo-XX/neo-XY sex-determining system has been found in 24 species (or 26, considering *Orthemis levis* and *Lestes vigilax*, see below), in one subspecies proposed by Kiauta [69] (*Crocothemis servilia mariannae* ssp. n.), in one or more populations of six species and in one or some individuals from a population of *Orthemis discolor* [6,53] (Table 2).

The multiple $X_1X_1X_2X_2/X_1X_2Y$ system is another derived sex-determining system present in *Micrathyria ungulata* [18] and probably in *Orthemis levis* and *Lestes vigilax* based on the presence of a trivalent in the individuals studied [5,31]. An alternative possibility is that the latter species have a neo-XY system and an autosomal trivalent, though no heterozygosity has ever been reported for an autosomal fusion or fragmentation.

## 5. Conclusions

The study of the meiosis and the characteristics of the heterochromatin in the X chromosome of three species of Gomphidae, together with data from the bibliography led us to propose that the ancestral diploid number of the family was 23 and that the unusually large size of the sex chromosome was due to an increase in heterochromatin rather than to structural rearrangements, as previously claimed. We also propose that the increase in diploid number in species with neo-XY systems in gomphids originated by mechanisms of autosomal fragmentation and X-autosome fusion. Moreover, the analysis of the neo-sex determining systems allowed us to pose that two other species of odonates that present trivalents could have a multiple X1X1X2X2/X1X2Y sex-determining system.

**Author Contributions:** Conceptualization, L.M.M. and S.S.A.; methodology, S.S.A. and P.J.R.; Investigation, S.S.A., P.J.R., M.F.F. and L.M.M.; writing—original draft preparation, L.M.M.; writing—review and editing, S.S.A., P.J.R., M.F.F. and L.M.M.; funding acquisition, L.M.M. All authors have read and agreed to the published version of the manuscript.

**Funding:** This research was supported by grants from the National Council of Scientific and Technological Research (CONICET), Multi-year Investigation Project (PIP) N° 11220200102115CO.

**Institutional Review Board Statement:** Not applicable.

**Data Availability Statement:** The data presented in this study are available on request from the corresponding author.

**Acknowledgments:** We would like to thank to J. Muzón for the identification of the specimens of *Phyllocycla propinqua* and to A. Rodrigues Capítulo for the identification of the specimens of *Phyllocycla* sp.

**Conflicts of Interest:** The authors declare no conflict of interest.

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
