# Peer review of "Meiotic Analysis of Gomphidae Species Sheds Light on the Large X Chromosome of the Family (Anisoptera, Odonata)"

_diversity, doi:10.3390/d14100874_

Round 1

Reviewer 1 Report

 REVIEW

The manuscript makes a contribution to cytogenetics of the order Odonata primarily by the fact that it attempts to reveal the relationship between increasing X-chromosome size and amount of heterochromatin

using the example of representatives of the large family Gomphidae. The evolution of sex chromosomes is known to be a gradual and continuing process, and the accumulation of repetitive sequences is a common feature of sex chromosomes in eukaryotic organisms. Thus, the problem the MS is devoted with is interesting and important.

However, the manuscript has quite a few shortcomings.

Below I will list the comments that I had while reading each section of the article.

Introduction

Paragraph 2. Here it is written that «Odonata exhibits (should be exhibit) some particular cytogenetic features, such as ……. post-reductional meiosis….» In no way can I agree with this statement. The authors refer to their own thirty year old article (Mola 1995) and do not mention the article by Nokkala et al. (2002), which, on the contrary, shows that male meiosis in Odonata is pre-reductional (thus, conventional) but not post-reductional (inverted). Considering that the article under review does not deal at all with the problem of meiosis, I recommend removing this part from the Introduction or, in any case, saying that the question of the type of meiosis has not been resolved and there is an alternative point of view. Somewhere, either in Introduction or in Material and Methods or in Results, it should be mentioned all three species were studied earlier, that is, their karyotypes were known. It should be noted what is new revealed for these species in the present work. Moreover, from Table 1, it follows that Phyllocycla sp. is the same Phyllocycla sp. as in Mola (2007). First, it should be noted that prompted the authors to study the same species, while not indicating that it is the same. Second. it is a pity that over the past years, it has not been possible to identify Phyllocycla sp. - an increase in the number of indeterminate species with known karyotypes does not benefit anyone (especially since nothing particularly interesting or unique was revealed). Discussion This important section of the article has almost no references to the original works in question. Apparently, the authors believed that this was not necessary, since all the mentioned species are listed in the table. But this is not true, since the reader should easily navigate what the authors report (whose data, when received, etc.)                Lines 27, 278, 346-348. “…large size [of the X-chromosome] might have been due to a progressive accumulation of repetitive DNA and heterochromatinisation and not to fusion as previously suggested. Who, besides Perepelov & Bugrov (whose hypothesis you rightly rejected) claimed this? It is necessary to either strengthen the arguments, or remove this ending. Small remarks. I consider it incorrect to use haploid set as, e.g. n=11+X0, since X0 is the sex chromosome formula XX/X(0), whereas haploid set is 11+X, which means n=12(11+X). The authors themselves use this correct style on the line 228! English needs revision. In some cases, the sentences are too long and the meaning of what was said escapes (e.g. lines 198-204, 227-231). Some other comments and remarks, which I have annotated in the attached file, should be corrected before publication. So, the article needs to be seriously revised in order to be recommended for publication.

Reviewer 2 Report

I have some comments on the MS. The first one is about the chromosome images, could you please provide better images? In the image 1a for example, it is very difficult to count the chromosomes. Also, can you include a scale bar? 

Table 1. It is mainly part from "Kuznetsova VG, Golub NV (2020) A checklist of chromosome numbers and a review of karyotype variation in Odonata of the world. Comparative Cytogenetics 14(4): 501-540. https://doi.org/10.3897/CompCytogen.v14.i4.57062". Could you please cite in the text that it is adapted from there but with information regarding the chromosome X length? 

Why you did not perform C-banding? If your main result is to say that the X chromosome is this big due to the heterochromatin, the best way to see it is with C-banding + giemnsa, please consider to perform this and include into the MS. 

In the table 2 you have some labels that are moved, please have a look and correct it. 

I think this work can have a lot of interest but it needs to be improved a bit.

Round 2

Reviewer 1 Report

Dear Authors, thank you for revising the MS and correcting those parts of the paper that caused my criticism. In my opinion, the MS in the present form can be accepted for publication. 

Author Response

Thank you very much for your comments that have allowed us to improve the manuscript.

Reviewer 2 Report

Nothing to say, thanks for address and try to answer my last concerns

Author Response

Thank you very much for your comments that have allowed us to improve the manuscript